# Real-Time Detection of Microplastics Using an AI Camera

**DOI:** 10.3390/s24134394

**Published:** 2024-07-06

**Authors:** Md Abdul Baset Sarker, Masudul H. Imtiaz, Thomas M. Holsen, Abul B. M. Baki

**Affiliations:** 1Electrical and Computer Engineering, Clarkson University, Potsdam, NY 13699, USA; sarkerm@clarkson.edu (M.A.B.S.); mimtiaz@clarkson.edu (M.H.I.); 2Civil and Environmental Engineering, Clarkson University, Potsdam, NY 13699, USA; tholsen@clarkson.edu

**Keywords:** artificial intelligence (AI), DeepSORT, environmental monitoring, freshwater ecosystems, machine vision, microplastics (MPs), object detection, underwater detection, YOLOv5

## Abstract

Microplastics (MPs, size ≤ 5 mm) have emerged as a significant worldwide concern, threatening marine and freshwater ecosystems, and the lack of MP detection technologies is notable. The main goal of this research is the development of a camera sensor for the detection of MPs and measuring their size and velocity while in motion. This study introduces a novel methodology involving computer vision and artificial intelligence (AI) for the detection of MPs. Three different camera systems, including fixed-focus 2D and autofocus (2D and 3D), were implemented and compared. A YOLOv5-based object detection model was used to detect MPs in the captured image. DeepSORT was then implemented for tracking MPs through consecutive images. In real-time testing in a laboratory flume setting, the precision in MP counting was found to be 97%, and during field testing in a local river, the precision was 96%. This study provides foundational insights into utilizing AI for detecting MPs in different environmental settings, contributing to more effective efforts and strategies for managing and mitigating MP pollution.

## 1. Introduction

Microplastics (MPs), small plastic particles ≤ 5 mm [millimeter] in size [1], are identified as emerging contaminants, given their widespread presence and potential harm to ecosystems and human health. MPs are persistent in terrestrial, aquatic, and marine environments and carry contaminants to many parts of the trophic food webs [2]. MPs are recognized as one of the major environmental concerns by the United Nations [3]. Projections by the World Economic Forum suggest that without decisive action, MP contamination levels will double by 2030, and the weight of plastics in the ocean will surpass that of fish by 2050 [4]. According to recent toxicological research [5,6], MPs may have harmful health consequences, including tissue inflammation, feeding disturbance, poor growth, developmental defects, and alterations in gene expression. Hence, it is imperative to develop a comprehensive understanding and appropriate management strategies to safeguard aquatic environments and the organisms that depend on them. An important step towards mitigation is accurately estimating MP concentrations in water bodies and evaluating their dynamics.

Despite MPs’ ubiquity, there is a lack of technologies for rapidly and accurately identifying and quantifying MPs in an aquatic environment [7]. The development of this technology is a significant scientific challenge because of the small size of MPs. Naked eye, microscopy, spectroscopy, and thermal analysis-based detection techniques have low accuracy and are time-consuming. Optical, laser-based, and scanning electron microscopy techniques are significant advancements in MP detection; however, they are obtrusive, bulky, and contain expensive subsystems [8,9]. In addition, these methods are laboratory-based and require the water samples to be carried from the water body to the controlled laboratory. Therefore, these methods are not suitable for in situ detection of MPs in water. An in situ and real-time non-destructive detection method with high speed, convenience, and acceptable accuracy potentially utilizing artificial intelligence (AI) [10,11] to quantify and track MPs is desired. The application of an AI-based approach (e.g., machine/deep learning) could help with the detection of various subgroups of MPs in environmental settings, yet its implementation in the field of MPs identification is still limited [12].

Traditional methods for detecting aquatic MPs are labor-intensive, time-consuming, costly, and do not provide in situ, real-time monitoring data. We aim to overcome this gap by harnessing recent strides in AI computing and advanced image processing technology. This novel system will continuously capture real-time water sample images while the AI algorithm processes the visual data with precision, enabling the identification, quantification, and tracking of MPs. The combining of computer vision with different technologies has recently been demonstrated [13,14], and vision-based underwater monitoring has been implemented in many research studies. However, vision-based underwater MP detection has not yet been demonstrated except for our recent work [15], where we detected and counted MPs from real-time video using a commercially available camera housed in a waterproof clear enclosure, submerged in a lab flume, and interfaced with a laptop computer.

To automatically detect MPs in an underwater environment using a computerized method, an object detection model is required. One of the popular object detection models is You Only Look Once (YOLO), which has been widely used since 2015 [16,17,18,19,20]. YOLOv5 is one of the most popular YOLO models, which was also used for underwater target detection [21], maritime object detection [22], underwater scallop recognition [23], and sea cucumber identification [24]. While such object detection is required for real-time MP detection, sorting algorithms are generally used for tracking or velocity calculation. Deep simple online real-time Tracking (DeepSORT) is one of the multi-object tracking (MOT) algorithms that has been used in underwater object tracking [25,26,27,28], vehicle tracking [29], unmanned vehicle tracking (UAV) [30], athletics [31], robotics [32], population tracking [33], and cotton seed count [34]. In our current study, we leveraged YOLOv5 and DeepSORT architecture together.

Therefore, for the first time, this study aims to demonstrate the following:(1)A noble AI-vision implementation for the accurate detection and tracking of underwater MPs.(2)A detailed methodology for MP counting, velocity calculation, size measurement, and path detection over a wide range of water velocities, and lighting conditions, with different camera setups.(3)A validation of this lab-developed sensor system both in the controlled flume and in a river system.(4)An annotated dataset of underwater MPs to facilitate further research.

## 2. Methodology and Materials

The proposed AI-vision system is configured with an advanced optical camera(s) interfaced with a laptop computer and light-emitting diode (LED) lights. It captures images at a rate of 60 frames per second, with a resolution of 1920 × 1080 pixels. In the process of systems validation, MPs were released from the water surface and allowed to transport/settle with the flowing water. The trajectories of the MPs were then captured by the submerged camera. A Python-based script was developed (flowchart in Figure 1) in such a way that when the camera captures the image, the image was passed through the object detection YOLOv5 model to count the number of MPs in the image. If any MP was detected, the pixel coordinates of the detected MP in the image (center coordinates, width, and height of the object in normalized pixels) are sent to the DeepSORT to track the movement of the MP. The DeepSORT assigned a unique identity to each MP detected and followed its movement across the frames. From two consecutive images, the instantaneous velocity was computed using the measured distance of the MP and its travel duration.

The velocity of MP was also compared with the water velocity, which was measured using a Vectrino Plus (Nortek) acoustic Doppler velocimeter (ADV). We placed an imaginary vertical reference line at one-quarter of the frame width. When detected MPs pass the reference line, we increased the count with respect to the DeepSORT assigned identity thus reducing errors arising from DeepSORT. In a controlled experimental setup, various features of the sensor (e.g., camera focus, focus length, and lighting) and water body (e.g., water velocity, depth, etc.) were tested to optimize the detection and tracking efficiency. The developed Python script allows us to configure important parameters, such as trained model weight and the confidence level of object detection models. Model weights are parameters that the algorithm learns from the training data. The confidence label of a model refers to the level of certainty or trustworthiness assigned to the predictions made by the model. Typically, lowering the confidence level increases false positives, while raising it decreases them.

### 2.1. Experimental Setup

The experiment was conducted within the recirculating open channel flume situated in the Water Resources Laboratory at Clarkson University, USA, measuring 12 m [meter] in length, 0.45 m in width, and 0.77 m in height. The side walls of the flume are transparent and have an adjusted bed slope of S_0_ = 0.5%. The flume’s spatial dimensions were defined along the longitudinal (x), transverse (y), and vertical (z) axes. An observation area measuring 2.00 × 0.46 m was designated at 5.0 m downstream from the flume inlet, chosen to ensure the establishment of fully developed turbulent flow conditions (shown in Figure 2). Flow rate measurements were obtained through pressure gauges interfaced with a computer system. To control flow depth, a point gauge and tailgate were employed at the terminal end of the flume.

A Vectrino Plus (Nortek) ADV was utilized to measure the time-averaged water velocity in the longitudinal direction. A funnel was affixed to the point gauge, and the funnel opening was 10 mm. The funnel helps to channel the MP towards the designated location with minimal distortion. The lower section of the funnel remained consistently positioned 2 cm below the water surface to mitigate interference from surface effects and turbulence. A white, waterproof, thin white sticker paper was used as a backdrop on the flume wall. After release, the MPs were captured using a fine mesh with a 0.5 mm aperture size located at the terminus of the flume. We conducted the experiment with three flow scenarios, where the time-averaged water velocities (*U*) were 15, 25, and 36 cm/s for offline video analysis and data collection. For real-time analysis, we utilized a total of four turbulent flow scenarios having time-averaged water velocities of 15, 25, 35, and 46 cm/s. For these flow scenarios, the Reynolds number (Re=ρUH/μ, where the average water depth is *H* (0.31 m), *ρ* is the water density, and *μ* is the dynamic viscosity of water) was in the range of 46,500 < *Re* < 142,600, indicating a fully turbulent flow (*Re* > 10^4^) in all the scenarios. To make the camera system waterproof, we placed the cameras into a waterproof IP67 polycarbonate submersible, see-through, lift-off enclosure [35]. The waterproof LED, SOLA Light and Motion 2500 [36], which has three brightness settings: high (2500 lumens), medium (1000 lumens), and low (500 lumens), was used. The coverage angle of each light was 60 degrees. For data processing, we used a Lenovo Legion 7 laptop with a Core i7-10750H CPU @ 2.60 GHz, NVIDIA GeForce RTX 2070 GPU, and 1 TB SSD and Ubuntu 22.04 as an operating system. The Python version was 3.10.

### 2.2. Camera Setup

Initially, a single-camera system was used; three different cameras were tested (Figure 3 and Table 1). Each camera setup (Table 1) consists of an enclosed camera and two LED waterproof lights. The sizes of the enclosures were 3 × 3 × 2, 4 × 2 × 1, and 4 × 2 × 1 inches. All these camera interfaces used USB for connection to the computer. For offline video analyses, we used 1920 × 1080 resolution, and for real-time analysis, we employed Camera 1 with a 1270 × 720 resolution (as explained in the next section).

### 2.3. Microplastics

We used different sizes, shapes, colors, materials, and densities of MPs (Table 2). The MPs were submerged in the experimental water for three hours before use to minimize static surface charges.

### 2.4. Dataset and Training

We collected 484 videos from the lab and the field for the training process. These videos cover eight different distances (190, 210, 230, 250, 270, 290, 310, and 330 mm) from the camera and four sizes (2, 3, 4, and 5 mm) of MPs. From the field, we have collected videos at 230 mm. We excluded videos in which MPs were not visible throughout the entire duration. Frames were extracted from the videos and resized to 1280 × 720 pixels. Next, we manually annotated 17,069 images and randomly divided them into groups of 80% for training, 10% for validation, and 10% for testing. The training process involved 300 epochs with a learning rate of 0.01. We used state-of-the-art YOLOv5n as the base model for transfer learning, as it is the smallest, so it has the fastest processing speed of all Yolo5 models.

### 2.5. Validation Study in Lab and Field

We performed offline and real-time video analysis in the lab. In the field, we performed offline video analysis. The camera was positioned outside the flume in the lab and submersed in the field, and the focus was calibrated for each distance in both scenarios. 

Lab study (offline). For offline analysis, MPs were released at varying distances from the camera (190, 210, 230, 250, 270, 290, 310, and 330 mm) at water velocities of 15, 25, and 36 cm/s. We captured videos at 1920 × 1080 resolution and 60 fps. Following the video recording, we assessed the detection accuracy of our model by comparing it with manually counting the released MPs, and Cameras 1, 2, and 3 were used. 

Lab study (real-time). We captured real-time video at 1280 × 720 resolution and a maximum frame rate of 28 fps. Individual MPs were released at water velocities of 15, 25, 36, and 46 cm/s, and the system performed real-time counting, size calculation, and velocity measurements. MPs were released at a fixed distance of 230 mm from the camera, and only Camera 1 was used.

Field study (offline). The field test was conducted in the Raquette River, Potsdam, New York (coordinates approximately 44.672820 latitude and −74.995604 longitude) using a custom deployable structure at 914 mm in height and 469 mm in width to accommodate all electronic and mechanical components to replicate the controlled flume environment within the flowing river (shown in Figure 4). It contained an adjustable vertical bar, allowing the Camera 1 setup to be positioned based on the river’s water depth.

Two horizontal wooden supports securely held a funnel resting upon a separate adjustable aluminum frame, enabling both horizontal and vertical positioning control and was used for MPs dispersal at any water depth. A white-colored plastic board was placed as a backdrop to make a uniform background. We conducted the experiment by releasing a single MP at a time upstream using the funnel located 2 cm below the water surface and 230 mm from the camera and repeated the process with four different MP sizes. The water depth was 44 cm. For each size of MPs, we released five particles at the same location. We also measured the water velocity using the handheld ADV (SonTek FlowTracker2) [40].

### 2.6. Dynamic Coverage Computation

We released MPs from N (N = 8) at different points on the water surface, i.e., varying distances from the camera (across the flume width). To accurately calculate the velocity (d) and size of the MPs at each distance, measurements for the width and height of camera coverage at that distance are required. Figure 5 depicts a scenario where O represents the camera position, with the camera coverage width and height denoted as W and H, respectively; note that both OP and OQ are perpendicular to AB and BC, respectively.

EF and FG, which are the width (W) and height (H) of the frame coverage, respectively. As OPB and OMF are right triangles, the ratios of corresponding sides in similar triangles OPB and OMF are calculated as follows:(1)$$MF=\frac{OM}{OP}\times PB$$
(2)W=2MF

Here, W is the width of the frame at a distance of *d*.

Similarly, OBQ and OFN are the right triangles. The ratios of corresponding sides in similar triangles OBQ and OFN are as follows:(3)$$NF=\frac{ON}{OQ}\;\times \;BQ$$
(4)H=2NF

Here, *H* is the height of the frame at a distance of d.

MP Velocity Computation. To compute the velocity of MPs, we first computed the horizontal ($v_{x}$) and vertical components ($v_{z}$) of the velocity and then the resultant velocity ($v_{T}$) in cm/s:(5)$$v_{x}=\frac{W\times \Delta xp}{Wxp\times \Delta t}$$
where $v_{x}$ is the velocity component of MP in horizontal (x) direction, W is the camera converge width at a distance *d* from the camera computed from Equation (2), and Δxp is the displacement of MPs in pixels, the displacement of MPs was measured in pixels based on their position in the previous frame, Wxp is the width of the frame in pixels, and Δt is the change in time in seconds, or the time it takes to move from its position in the previous frame.
(6)$$v_{z}=\frac{H\times \Delta zp}{Hzp\times \Delta t}$$
where $v_{z}$ is the MP velocity, H is the camera converge height at a distance *d* from the camera computed from Equation (4), Δxp is the displacement of MPs in pixels, and Hzp is the height of the frame in pixels. 

After computing the horizontal velocity $v_{x}$ component, a sliding window average was employed over a specified window size *k*; in our case, *k* = 5. If we denote the input sequence as the equation for the sliding window average (*SWA*) at each position, *i* is given by the following:(7)$$v_{xi}=\frac{1}{k}\sum_{j=i-k+1}^{i}v_{xj}$$
where, $v_{xi}$ is the sliding window average of the horizontal velocity at position i. Similarly, for the vertical velocity,
(8)$$v_{zi}=\frac{1}{k}\sum_{j=i-k+1}^{i}v_{zj}$$
where, $v_{zi}$ is the sliding window average of horizontal velocity at the position i.

The resultant instantaneous velocity $v_{T}$ can be obtained from the following equation:(9)$$v_{T=\surd (v_{xi}^{2}+v_{zi}^{2})}$$

The average velocity $v_{T(avg)}$ for N data points can be obtained using the following.
(10)$$v_{T(avg)}=\frac{1}{N}\sum_{i=1}^{N}v_{T}$$

Size computation. The equation below was used to calculate the size of MPs.
(11)$$S=\frac{W\times Dxmp}{Wp}$$
where Dxmp is the diameter of the MPs in pixels, Wp is the width of the camera frame in pixels, and W is the width of the coverage width of the frame. 

After obtaining the size of the MPs in a frame position, the sliding window average was implemented over a specified window size *k*. We denoted the input sequence using the equation for the sliding window average (*SWA*) at each position *i* as given by the following:(12)$$S_{i}=\frac{1}{k}\sum_{j=i-k+1}^{k}s_{j}$$

$S_{i}$ is the sliding window average at the position i and k is the size of the sliding window; in our case, *k* = 5. The window includes the current element *S_i_* and the *k* − 1 preceding elements.

Finally, we obtained the average size of the object:(13)$$S_{avg}=\frac{1}{N}\sum_{i=1}^{N}s_{i}$$
where *N* is the total number of data points, and *St* represents the size at time *t*. 

Precision calculation. Precision is the ratio of correctly predicted positive observations to the total predicted positives. It was calculated using the following formula:(14)$$Precision=\frac{True\;Positive\;(TP)}{True\;Positive\;\left(TP\right)+Falase\;Positive\;(FP)}$$

## 3. Results

### 3.1. Offline Video Analysis Results

The average precision of Cameras 1, 2, and 3 was 91%, 89%, and 87%, respectively, in offline testing (Figure 6). While a higher megapixel (MP) count proved helpful, we also found that the quality of the camera sensor played a crucial role. Surprisingly, Camera 2, at only 2 MP, performed similarly to the 16 MP Camera 3, emphasizing the importance of camera quality over sensor pixel count. All camera setups featured auto-brightness control, but only Camera 2 and Camera 3 had an autofocus capability. The intricate interplay between camera quality, including megapixel count and sensor quality, and environmental factors, such as light angle and brightness, was evident.

During testing, both offline and real-time analysis revealed that the confidence level of the object detection model plays a crucial role in controlling the percentage of false positives. When the confidence level is excessively high (approaching 100), false positives decrease, but the tracking performance of the DeepSORT model diminishes. Conversely, when the confidence level is set too low, DeepSORT tracking performs well, but false positives increase. Therefore, it is essential to set the appropriate confidence level based on the number of objects present in the water. In our study, we found that our model performs best within the 40% to 70% range of object detection confidence levels.

Particles being closer to the camera increased detection accuracy. However, this benefit comes at a cost since the closer the MP is to the camera, the shorter the time it is visible while moving. A distance closer than 190 mm from the camera or water velocities higher than 46 cm/s reduced counting accuracy. As expected, being too far (more than 330 mm) from the camera also reduced the detection capability. In our camera setup MP detection worked best from 190 mm to 230 mm distance. 

There was a close correspondence between the calculated sizes by the system and the actual sizes of the MPs (average error of 5.5%) (Figure 7). Note that the MP size calculation was based on one fixed distance from the camera. However, the actual measuring points along the entire path varied and increased with an increasing Reynolds number due to an increase in turbulence (*Re* varied from 46,500 to 111,600).

The velocity variances for each type of MP for the three camera setups and eight lateral distances of MP dropping points are shown in (Figure 8). At each distance, the variance was calculated using the mean velocity measured by the three cameras, assuming a constant MP distance from the camera. The average variance in velocity at 15 cm/s was 0.62 ± 0.32 cm/s (avg ± std), at 25 cm/s it was 1.6 ± 0.85 cm/s, and at 36 cm/s it was 2.8 ± 1.5 cm/s. The MPs measured velocity variance increased at higher water velocity due to the turbulence and travel path deviation due to turbulence, shape, density, and other environmental factors.

### 3.2. Field Test Result (Offline)

Camera 1 yielded the best performance (91%) in our offline lab tests; therefore, it was used for field tests in which MPs were released at a 230 mm distance from the camera and the water velocity was 5.0 cm/s. For these tests, the average measurement precision was 96%, with only one false positive (natural particle detected as MP) for the 4 mm MPs (Table 3). In the laboratory experiment conducted at a distance of 230 mm and a water velocity of 15 cm/s, the average precision was 91% using Camera 1. However, in the field when the water velocity was only 5 cm/s, there was an improvement in precision of 6%. This improvement was likely because at lower water velocity, more frames could be analyzed, and in addition, the water in the field was clearer than in the lab flume.

### 3.3. Real-Time Analysis Result (Lab)

The average precision for all sizes at different water velocities was 97% for real-time detection (Figure 9). The variance gradually increased at higher water velocities (Figure 9b) and was generally larger than found when analyzing videos offline due to higher velocities and lower image resolution. The variance in velocity at 15 cm/s was 10 ± 0.36 cm/s (avg ± std), at 25 cm/s was 14 ± 1.9 cm/s, at 35 cm/s was 22 ± 2.3 cm/s, at 46 cm/s was 21 ± 3.1 cm/s. It is evident that with the increase in water velocity, the calculated variance increases. This difference was related to the difference in image resolution since at lower resolution, the distance represented by each pixel increases. For Camera 1, at a distance of 230 mm, the per-pixel distance is 0.13 mm for 1280 pixels and 0.086 mm for 1920 pixels. Consequently, even a small shift in the detected center of MPs can lead to a significant increase in variance. While higher-resolution images provide better MP detection, they also require more processing power. In real-time detection scenarios, particularly with high water velocities, faster processing is crucial. Therefore, to maintain performance with our current model at high water velocity, we reduced the image resolution from 1920 × 1080 pixels to 1280 × 720 pixels. Size estimation during real-time testing was similar to offline analysis, with average size calculation errors of approximately 10%.

## 4. Discussion

The deployment of three distinct camera setups revealed that the Camera 1 configuration yielded the best performance. During real-time detection, this system maintained precision even under increasing water velocity and turbulence. However, this finding highlights a challenge: the variance in MPs’ measured velocity is influenced by water dynamics and the MPs’ physical properties since a constant distance from the MP to the camera is assumed. Using a 3D camera would overcome this issue, because of its enhanced depth perception, which would increase velocity measurement accuracy. However, in our experiments, the 3D camera did not work well. According to the manufacturer, this camera model should provide distance calculation at distances greater than 300 mm. In our case, the flume wall was 450 mm away, however only reflections of the background were seen. While autofocus cameras can offer advantages in focus accuracy and adaptability to diverse applications, this can be offset by their increased complexity and slower performance compared to fixed-focus cameras. In addition, as an autofocus camera continuously adjusts its focus, it consumes more power compared to fixed-focus cameras.

### 4.1. Experimental Conditions

To achieve useful velocity measurement data, precise MP release positioning is crucial due to the turbulence generated by MPs entering the water. In our experiments, a horizontally and vertically adjustable funnel was used. To prevent adherence of MPs to the funnel, a spoon containing a small amount of water was used during the MP release process. The opening of the funnel was small to eliminate turbulence created by the funnel itself. A large opening shifts the dropping point, making the measurement unstable. 

DeepSORT is designed to maintain the identity of individual objects between video frames. However, the MPs can be very small and occupy only a few pixels. This limited size results in fewer features being extractable, leading to increased identification errors in DeepSORT and inaccurate counting. To mitigate this problem, we used a virtual vertical reference line placed at one-quarter of the frame width and incremented the count each time a microplastic crossed this line. 

To accurately calculate the actual size and velocity of MPs, precise annotation as close as possible to the edge of the MPs is crucial. If the provided annotations for training are inaccurate, it negatively impacts the precision of the size calculations. In our case, annotating the 2 mm MPs proved challenging due to their small size, making it difficult to achieve exact annotations consistently. During annotation, we observed instances where the bounding box size exceeded the actual MP size, leading to inflated size predictions.

Light angle and brightness play a vital role in the detection of MPs because of water’s unique optical properties, including refraction, backscatter, and color absorption, which necessitate a strategic approach to lighting. We found that ambient light caused issues with detecting MPs in both flume and field tests. In the lab environment, we attempted to reduce ambient light in the flume by turning off other light sources. In the field, where we only tested at 44 cm water depth, we blocked the sunlight from entering the viewing field using a black cloth. 

While we have yet to conduct extensive tests in seawater, the system is expected to perform similarly, provided that adjustments are made for the different optical properties and potentially higher turbidity levels in marine environments. The system is designed to calculate velocity and track MPs while the camera is stationary. This method can be adapted for use on a ship in both ocean and river environments by mounting the camera and lighting systems on a stabilized platform and using real-time image processing capabilities. By incorporating the moving platform’s velocity into the equation, this system can be modified for use on moving ships. Future studies will focus on validating the system in various marine settings to confirm its robustness and adaptability.

### 4.2. Camera Limitations

There were some limitations to the camera setup used; the LED had an illumination angle of only 60 degrees, which is insufficient for imaging moving MPs. To produce equal illumination, less sharp shadows, and soft, diffused lighting, a wide-angle light source would be better. This type of lighting is especially useful for small object detection, emphasizing the characteristics of the object and reducing errors. Additionally, by reducing specular reflections on bright surfaces, a wide light source can allow more object details to be seen. While our system performs well for particles from 2 to 5 mm in clear water conditions, the accuracy of detection decreased in darker water due to lower contrast between MPs and the background. Increasing the light intensity (e.g., using a stronger light source) and potentially changing the camera (e.g., to one with higher sensitivity in low light conditions) could mitigate this issue. The system we developed was trained on commercially available MPs of various sizes and colors. While the system has performed well, implementing it in real-world scenarios will require collecting and annotating a large amount of data featuring MPs from natural sources to achieve optimal detection and counting accuracy. 

### 4.3. Future Work

In the future, 3D camera technology will be incorporated into the existing system to enable the determination of the distance of the MPs in each position, rather than calculating velocity using constant distance, thereby reducing the unwanted variance in velocity calculation. In addition, a more lightweight deep learning model for MP detection needs to be developed to increase efficiency and resource optimization and enable deployment on resource-constrained platforms such as Jetson Nano, Google Coral, etc. To make a robust system that can differentiate between natural particles from the MPs, we will collect a large amount of data from natural rivers and seas to train the model. We will explore the feasibility and advantages of implementing the entire MP detection system on field-programmable gate array (FPGA) hardware and investigate how FPGA can contribute to real-time processing, reduced power consumption, and increased adaptability for deployment in diverse underwater environments. 

## 5. Conclusions

This study presents a comprehensive framework for measuring MPs in aquatic ecosystems in real-time. By combining deep learning-based MP detection and advanced object-tracking algorithms, this study contributes to the development of efficient and accurate methods for enabling functionalities like counting, velocity calculation, size measurement, and path detection across diverse setups. After extensive and careful testing of three different camera configurations, Camera 1 (16 MP fixed-focused from econ-Systems) was the most efficient and dependable. The system was optimized for distances ranging from 190 mm to 330 mm from the camera. Within this range, the system demonstrated high detection accuracy, with precision rates of 97% in laboratory settings and 96% in field tests for MP sizes from 2 mm to 5 mm. Detection performance decreased significantly for distances closer than 190 mm or farther than 330 mm due to either insufficient visibility time or reduced detection capability. Our findings underscore the feasibility and effectiveness of employing fixed-focus-based camera systems for future research and monitoring efforts in this domain. The successful implementation of our model in both controlled lab settings and real-world scenarios shows its practical applicability and potential for broader adoption in tackling the pervasive issue of MP pollution in aquatic ecosystems. 

## Figures and Tables

**Figure 1 sensors-24-04394-f001:**
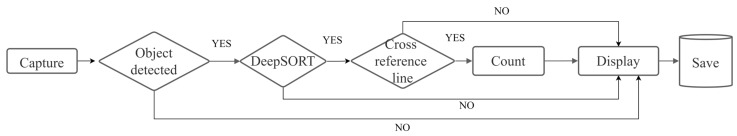
Firmware flowchart of the proposed system.

**Figure 2 sensors-24-04394-f002:**
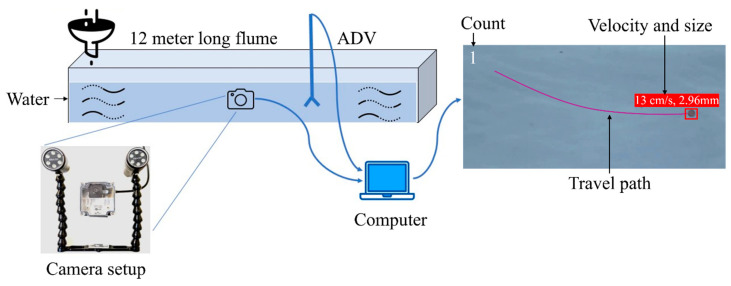
Controlled laboratory experimental setup for detection of MP in open channel flume; acoustic Doppler velocimeter (ADV).

**Figure 3 sensors-24-04394-f003:**
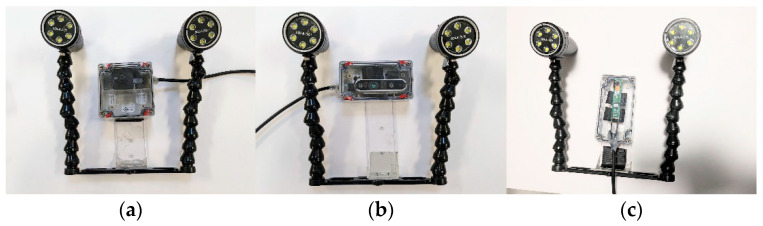
Different cameras were employed in this study: (**a**) Camera 1, (**b**) Camera 2, and (**c**) Camera 3.

**Figure 4 sensors-24-04394-f004:**
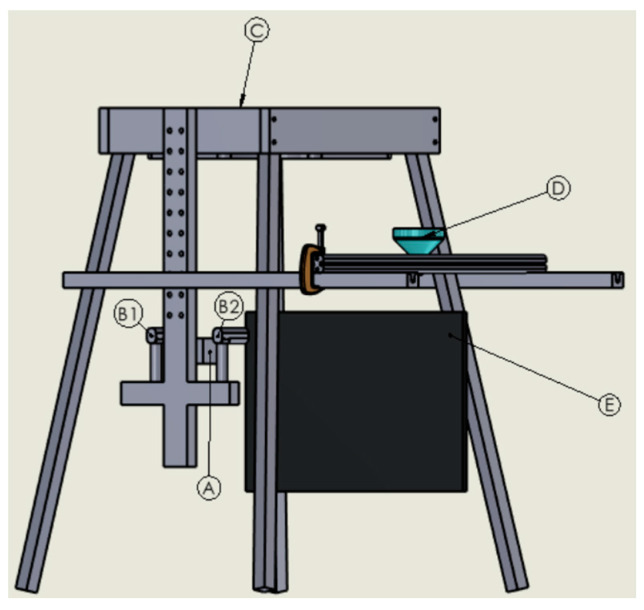
Experimental setup for the field test (A) Camera support, (B1, B2) LEDs, (C) laptop holder, (D) funnel, and (E) backdrop.

**Figure 5 sensors-24-04394-f005:**
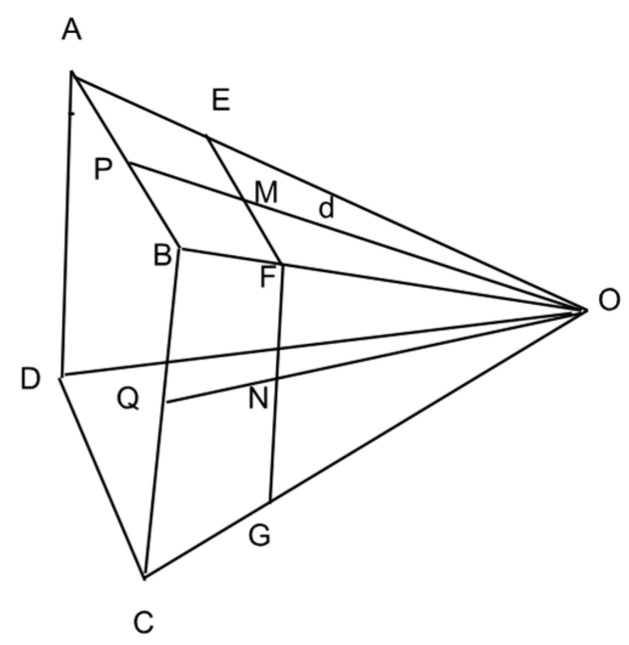
Illustration of camera coverage plane for different distances from the camera.

**Figure 6 sensors-24-04394-f006:**
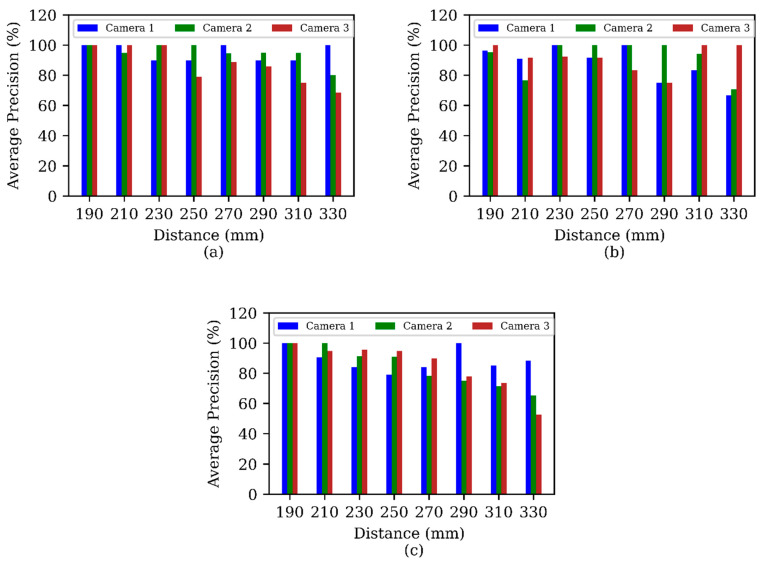
Comparison of MPs’ count detection average precision among three cameras (Camera 1, 2, and 3) for (**a**) underwater velocity of 15 cm/s, (**b**) 25 cm/s, and (**c**) 36 cm/s.

**Figure 7 sensors-24-04394-f007:**
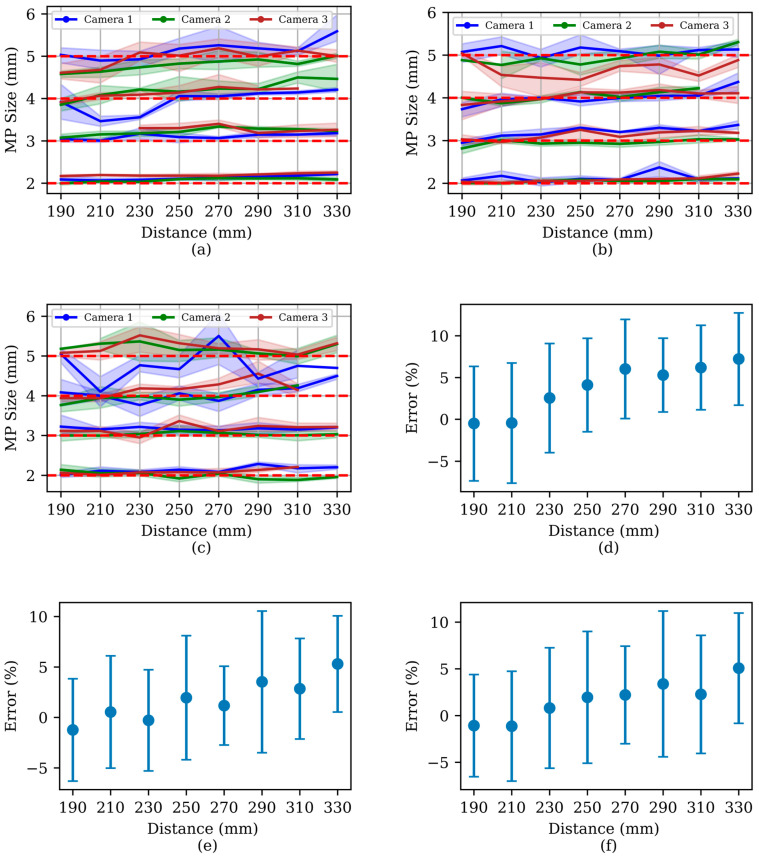
Comparison of MPs’ size computation errors. Among three cameras (Camera 1, 2, and 3) for underwater velocity of (**a**) 15 cm/s, (**b**) 25 cm/s, and (**c**) 36 cm/s (red dotted lines are the actual size of MPs) and MPs’ size computational error at Reynolds number (**d**) 46,500 (**e**) 77,500, and (**f**) 111,600 for underwater velocities of 15, 25, and 36 cm/s, respectively.

**Figure 8 sensors-24-04394-f008:**
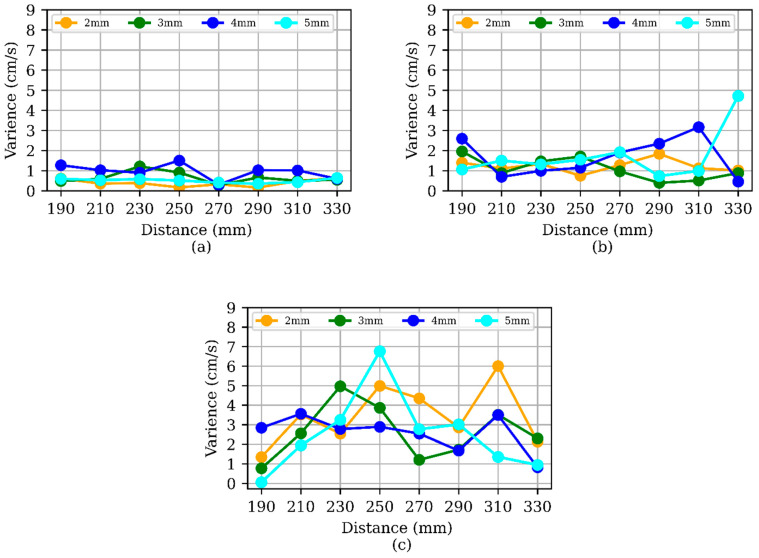
Comparison of MP’s velocity variance at (**a**) 15 cm/s water velocity, (**b**) 25 cm/s water velocity, and (**c**) 36 cm/s water velocity.

**Figure 9 sensors-24-04394-f009:**
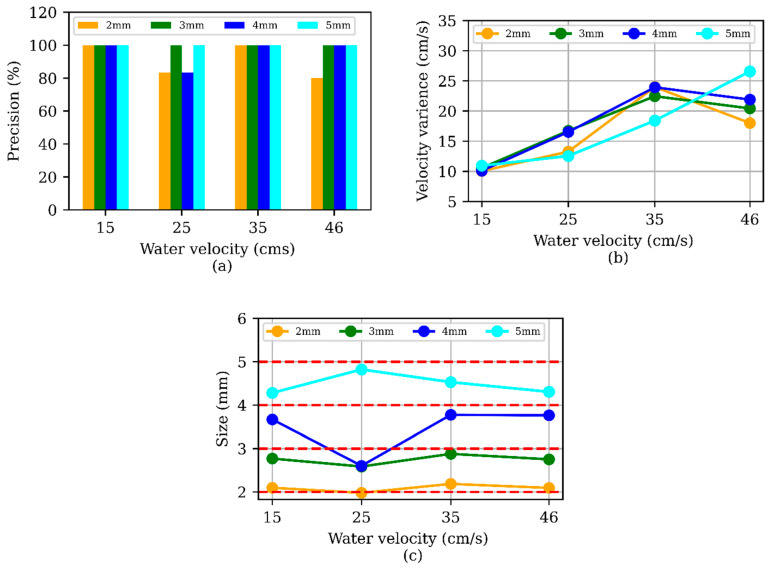
(**a**) Precision change over water velocity for real-time test, (**b**) velocity variance over different water velocities, and (**c**) size detection in real-time analysis (red dotted lines are the actual size of MPs).

**Table 1 sensors-24-04394-t001:** Comparison of different camera specifications.

Features	Camera 1	Camera 2	Camera 3
Manufacturer	See3_Cam [37]	Intel RealSense [38]	See3_Cam [39]
Type	2D	3D	2D
Sensor Resolution	13 MP	2 MP	13 MP
Focus	Fixed	Autofocus	Autofocus
Brightness Control	Auto	Auto	Auto
FPS	60 fps@1920 × 1080 120 fps@640 × 480	30 fps@1920 × 1080	30 fps@1920 × 1080
Interface	USB 3.1	USB 3.0	USB 3.1

**Table 2 sensors-24-04394-t002:** The types and sizes of MPs used in this study.

Size (mm)	Shape	Actual Size (mm)	Color	Polymer Type	Density (kg/m^3^)
5 mm	Spherical	4.96	Cyan	Polystyrene	1050
4 mm	Spherical	3.98	White	Cellulose Acetate	1300
3 mm	Spherical	2.96	Green	Acrylic	1190
2 mm	Spherical	1.98	Orange	Cellulose Acetate	1300

**Table 3 sensors-24-04394-t003:** Field test result for Camera 1.

Size	Actual Count	Software Count	Precision (%)
2 mm	5	5	100
3 mm	5	5	100
4 mm	5	6	83
5 mm	5	5	100

## Data Availability

All data assembled or analyzed in this study are available from the corresponding author.

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
