# Peer review of "Real-Time Detection of Microplastics Using an AI Camera"

_sensors, 2024, doi:10.3390/s24134394_

Round 1

Reviewer 1 Report

Comments and Suggestions for Authors

1. How does this article address the technical shortcomings of quickly and accurately identifying and measuring MPs in aquatic environments?

2. What are the advantages of this method compared with existing methods?

3. Your tables should be more standardized.

4. You should conduct a detailed analysis of your experimental results to verify the validity of the methods presented in this paper.

5. 83% of the items in Table 3 should not have a % sign.

Author Response

Dear reviewer, 

The authors would like to thank the reviewer for his/her interest and careful review of the manuscript and for insightful and thought-provoking comments and recommendations. We have carefully gone over the manuscript and incorporated these suggestions into the revised manuscript.

We have added the response to the pdf file

Reviewer 2 Report

Comments and Suggestions for Authors

Review of manuscript

AI-Vision based Camera Sensor for Real-time Underwater Microplastics Detection by Md Abdul Baset Sarker, Masudul H. Imtiaz, Thomas M. Holsen, Abul B.M. Baki

I generally like the paper because it provides a method for detecting microplastic pollution and recommend publication after some minor revision.

(1) I do not appreciate hyphenation and abbreviations in the title of the paper. I also recommend changing the title as Camera Sensor Based on Artificial Intelligence Vision for Real-Time |Detecting of Microplastic Pollution in the Sea

(2) Indicate the country in addition to the Clarkson University

(3) To many sentences in the abstract describing the importance of the problem and poor existing methods for detecting pollution as well as advertising the new method suggested by the authors and high potential of this study. Just starts that: “The main goal of this research…..” and then describe the novelty you suggest including short description of the cameras and model.

(4) Please clearly indicate whether this method can be applied from a moving ship in the ocean or river and how this can be achieved. Thus, you will attract more readers than in the case of pure laboratory testing.

(5) Please exclude the second paragraph “In the future 3D camera technology will be incorporated…“ from the text or replace it. These are your plans not conclusions. Add more real informative summary in this section.

Comments on the Quality of English Language

English is OK

Author Response

(The authors gave the same response as above.)

Reviewer 3 Report

Comments and Suggestions for Authors

This article is a good initiative of Underwater Microplastics Detection by AI-vision based camera. I have some observation about this article.

Authors should clearly describe their dataset.

What is the range of the detection?

What are the influences of lighting conditions on underwater imaging?

What is the performance of detection in sea or river water?

Author Response

(The authors gave the same response as above.)

Round 2

Reviewer 1 Report

Comments and Suggestions for Authors

Accept.